# An International Study of Correlates of Women's Positive Body Image

Sandra Torres [1,*], Carolina A. Araújo [1], Amanda Fitzgerald [2], Barbara Dooley [2], Angeliki Leondari [3], Cátia Miriam Costa [4], Dorit Olenik-Shemesh [5], Efthymia Sygkollitou [6], Josip Burusic [7], Liesbet Boone [8], Marijana Šuvak-Martinović [9], Maritta Välimäki [10,11], Minna Anttila [11], Tali Heiman [5] and Toni Babarović [7]

[1] Faculty of Psychology and Education Sciences, Center for Psychology at University of Porto, University of Porto, 4099-002 Porto, Portugal
[2] School of Psychology, University College Dublin, D04 V1W8 Dublin, Ireland
[3] Faculty of Human Sciences, University of Thessaly, 382 21 Volos, Greece
[4] Instituto Universitário de Lisboa (ISCTE-IUL), Centro de Estudos Internacionais, 1649-026 Lisboa, Portugal
[5] Department of Education and Psychology, The Open University of Israel, Ra'anana 4353701, Israel
[6] Department of Psychology, Aristotle University of Thessaloniki, 541 24 Thessaloniki, Greece
[7] Ivo Pilar Institute of Social Sciences, 10000 Zagreb, Croatia
[8] Department of Developmental, Personal and Social Psychology, Ghent University, 9000 Ghent, Belgium
[9] Department of Psychology, Faculty of Humanities and Social Sciences, University of Mostar, 88000 Mostar, Bosnia and Herzegovina
[10] Xiangya Nursing School, Central South University, Changsha 410013, China
[11] Department of Nursing Science, University of Turku, 20014 Turku, Finland
[*] Correspondence: storres@fpce.up.pt

**Abstract:** Positive body image (PBI) has received attention in the recent research literature. Despite this, its role in daily functioning in different cultural contexts, particularly its potential relationship with academic outcomes, is still lacking. This study aimed to offer an international perspective on the association between PBI and body mass index (BMI), perceived academic achievement, and educational aspirations, as well as the mediating role of self-esteem. A cross-national study was conducted in eight European countries with a total of 2653 female university students. Participants completed an online survey measuring PBI (conceptualized as body appreciation), self-esteem, perceived academic achievement and aspirations, and body mass index (BMI). Results revealed differences in PBI between countries (low magnitude). PBI correlated negatively with BMI in all national groups (low-to-moderate magnitude). Mediation analysis showed that self-esteem mediated the association between PBI and academic variables. Findings from this study suggest that building students' self-esteem and PBI can be a suitable way to boost academic success.

**Keywords:** positive body image; academic achievement; educational aspirations; self-esteem; female university students

## 1. Introduction

Body image plays a vital role in several areas of people's life [1]. The negative consequences of poor body image cut across all key areas of living, including psychological and physical functioning [2]. Extensive research relates poor body image to low self-esteem [3,4], symptoms of depression [5,6], and negative health outcomes, such as unhealthy eating and lower levels of physical activity [7]. In addition to these well-established harmful effects, body image problems have also been hypothesized to undermine academic achievement and educational/career aspirations, e.g., [8–11]. According to Yanover and Thompson [11], body image dissatisfaction can lead to a high level of class absenteeism due to social anxiety regarding appearance. Concerns about one's look, size, weight, and shape can also inhibit attentional processes, such as concentration [11], diminish confidence [12], and, therefore, performance. Determining the specific role that body image plays in academic achievement

during the school years is a topic of interest since academic performance seems to be a strong predictor of social and occupational success in adult life [13]. Academic achievement is an important determinant of good job opportunities, desired earnings, and higher social status [10].

In this research field, we find studies that investigate the role of body image in academic performance, e.g., [8,14], while other studies place emphasis on weight status, mostly examining whether overweight and obesity disrupts academic functioning. The basic assumption is that excess weight can be related to lower academic achievement due to the stigmatized nature of obesity [15] and the possible impact of obesity on cognitive functioning [16]. A recent systematic review concluded that there is insufficient evidence to support a direct link between obesity and poor academic performance in students under 20 years of age [17]. Some authors have argued that the perception of being overweight, more than one's real weight status, could be a key factor in understanding the possible influence of body image on academic achievement [18,19]. In addition, it is also plausible that causal processes linking academic performance and weight may differ according to the development stage [17]. Substantially fewer studies have been conducted with emerging adults [20], who are identified as a unique life stage [21], thus limiting researchers from drawing robust conclusions.

Compared with weight condition, far less attention has been paid to body image experience and its influence on academic success. Studies with middle (13–14 years old) and high school students (15–18 years old) have observed an association between body dissatisfaction and poor academic achievements, mainly due to school avoidance and lack of participation in classes [12,22,23]. However, divergent results also exist. In college/university students, the study by Elsherif and Abdelraof [14] did not find a significant correlation between body dissatisfaction and academic behavior scores that were indicative of academic achievement. However, other investigations have documented that body dissatisfaction and body concerns were correlated with poorer academic achievement and functioning [8,11,24], possibly as a consequence of absenteeism and social anxiety about one's physical appearance [11]. On the other hand, higher academic achievements have been observed in students who were able to reject appearance norms [24] or were less focused on their physical appearance [10,25].

Globally, there are mixed results from studies analyzing the association between body image and academic success in all educational stages. Differences between study findings can be due to methodological heterogeneity in the assessment of academic achievement, considering that several indicators have been used, such as grade point averages (GPAs; e.g., [15,23]); intelligence quotient, e.g., [26]; school attendance, e.g., [27]; participation in class, e.g., [24]; and school functioning, e.g., [28]. Most studies used "direct" measures of student performance, and scant attention has been paid to students' own perceptions of their learning, referred to as "indirect" measures of student performance. "Indirect" measures capture the psychological processes driving learning [29] and subjective perspectives of academic achievement and, therefore, are important tools to improve knowledge in this field.

In respect of the association between body image and educational and career aspirations, the current body of research is scarce and inconsistent. Some data suggested that negative body image could lead to lower expectations for a successful career for girls/women [30] as a consequence of decreasing participation in education (e.g., avoiding any involvement in school activities) and job opportunities (e.g., absenteeism, not going to job interviews) whenever they feel bad about how they looked [12,22]. In turn, positive attitudes toward body image may be associated with greater career decision-making self-efficacy [31]. In addition, there is some evidence that women who pursue, or plan to, a post-graduate education tend to focus less on physical appearance [25] and reveal a lower preference for thin body-ideal messages [32]. Altogether, these studies suggest that females high on educational/career aspirations report greater body-protective attitudes. Conversely, there are also data documenting that undergraduate females with higher edu-

cational/career aspirations report greater body dissatisfaction after being exposed to thin targets [33].

In sum, the relationship between body image and academic success is still under-researched and inconclusive. Whether body experience interacts with academic achievement and educational aspirations remains an open question. To increase knowledge on this topic, more studies are needed with larger samples [12] and from different cultural backgrounds, assuming that human behavior, culture, and self-definition are inextricably interwoven [34]. Furthermore, the study of body image should encompass a broader exploration of the concept, covering positive body image (PBI). This recent construct represents an important shift from a primary focus on body disturbances and extends beyond appearance satisfaction [35]. As a multifaceted construct, PBI can be defined as a sense of love and respect for one's body, containing aspects of body appreciation and acceptance, body image flexibility, a broad conceptualization of beauty, inner positivity, and body functionality orientation [35,36]. Body appreciation—one of the main facets of PBI—is positively correlated with various well-being indices, such as self-esteem, e.g., [37,38]; and life satisfaction, e.g., [38,39]. This potential protective role for emotional and psychosocial problems [40] makes the PBI concept an important variable to consider in the study of educational achievement and aspirations.

To fulfill the existing knowledge gap, we sought to examine PBI in a cross-cultural context by using a large and diverse sample of university students from eight countries. The purpose of this study was twofold. First, as empirical evidence for a negative association between body appreciation and BMI is mixed and a recent meta-analysis suggested that culture may play a moderator effect [41], we aimed to clarify the relationship between PBI and BMI. Second, we sought to analyze the association between PBI and perceived academic achievement and educational aspirations among female university students. To address this goal, we took into consideration the possible role of self-esteem in the association between these variables. Self-esteem, in particular, is an important variable to consider given its link with both academic achievement, e.g., [42–44]; and PBI, e.g., [5,45,46], in university students. Furthermore, self-esteem affects a person's thoughts, emotions, desires, and goals [47]. It has been shown that low self-esteem potentially leads to self-doubt and depression [48] and, in turn, high self-esteem facilitates self-confidence, good stress management, and coping with difficult situations [49]. Empirical evidence from longitudinal research also suggests that self-esteem can enhance academic success by supporting the development of personal resources and resilience competencies [50]. Thus, based on these theoretical and empirical arguments, we tested whether self-esteem explained (i.e., mediated) the relationship between PBI and educational variables, as previously hypothesized by Elsherif and Abdelraof [14].

Given that PBI has been identified as a significant positive construct to study in the field of body image [51], elucidating its possible role in an educational setting is important in extending our knowledge. Furthermore, identifying whether a relationship between PBI and academic success exists in university settings could be beneficial to professionals involved in creating supportive educational environments where all students can succeed [25,52].

## 2. Materials and Methods

### 2.1. Participants

For this cross-sectional study, a minimum sample size of 984 participants (123 per country) was set based on a power analysis for sample size calculation [53]: statistical power $(1 - \beta)$ was set at 0.95 ($\alpha = 0.05$), with small to medium effect size. The sample consisted of 2653 female university students, aged between 18 and 30 years ($M = 21.61$, $SD = 3.17$), living in eight different countries: Bosnia and Herzegovina ($n = 248$; $M = 22.1$ years, $SD = 2.17$), Belgium ($n = 135$; $M = 19.3$ years, $SD = 1.69$), Croatia ($n = 201$; $M = 22.2$ years, $SD = 2.22$), Finland ($n = 263$; $M = 23.9$ years, $SD = 3.50$), Greece ($n = 1040$; $M = 21.2$ years, $SD = 3.20$), Ireland ($n = 91$; $M = 20.2$ years, $SD = 2.36$), Israel ($n = 158$; $M = 25.7$ years, $SD = 2.63$),

and Portugal (n = 517; *M* = 21.2 years, *SD* = 3.13). Participating countries represented the countries from a working group formed as part of the European Cooperation in Science and Technology (COST) Action IS1210 Appearance Matters. Age (18–30 years old) was the only inclusion criterion. Most participants were undergraduate students (n = 1866; 70.3%), followed by graduate (n = 683; 25.7%), postgraduate students (n = 55; 2.1%), and other (n = 49; 1.9%).

### *2.2. Measures*

#### 2.2.1. Demographic Information

Participants provided demographic details consisting of age, country, height (in centimeters), and weight (in kilograms). Self-reported height and weight were then used to calculate BMI (kg/m$^2$).

#### 2.2.2. Positive Body Image

The Body Appreciation Scale-2 (BAS-2; [5]) assesses an overall positive orientation to the body and provides the closest and most precise measurement of a core PBI construct [52]. This instrument comprises 10 items (e.g., "I appreciate the different and unique characteristics of my body"; "I am attentive to my body's needs") rated on a 5-point Likert-type scale (1 = never, 5 = always). Higher scores indicate greater body appreciation. The BAS-2 demonstrated good internal consistency in samples of college/university students (Cronbach's $\alpha \geq 0.89$; [5,46]). In the present study, Cronbach's $\alpha$ varied from 0.91 to 0.96.

#### 2.2.3. Perceived Academic Achievement

The Self-Reported Study Performance (SRSP) is a 6-item measure that represents subjective parameters of academic achievement at university [54]. The anchors on the 5-point Likert scale varied according to the item. For example: "How satisfied are you with your grades at the university?" (1 = very dissatisfied, 5 = very satisfied); "Compared to the other students in your study program, how successful a student are you?" (1 = one of the worst students, 5 = one of the best students); "How important is it for you to attain high achievement during your study at university?" (1 = not important at all, 5 = very important). The SRSP showed good internal reliability (Cronbach's $\alpha = 0.84$) in a sample of high school students [55]. Cronbach's $\alpha$ in our study was from 0.85 to 0.89.

#### 2.2.4. Educational Aspirations

Educational aspirations were assessed by the Career Aspiration Scale-Revised (CAS-R; [56])—a 24-item 5-point Likert scale ranging from 0 (not at all true of me) to 4 (very true of me) that includes three subscales: Achievement Aspiration, Educational Aspiration, and Leadership Aspiration. Only the Educational Aspiration subscale (CAS-R/Ed) was used in this study. This 8-item subscale assesses the degree to which individuals aspire to have more competencies, training, or advanced education within their career (e.g., "I plan to reach the highest level of education in my field"; "I would pursue an advanced education program to gain specialized knowledge in my field"). This subscale revealed good internal reliability in a sample of graduate female students (Cronbach's $\alpha = 0.85$; [56]). Here, Cronbach's $\alpha$ was from 0.66 to 0.93.

#### 2.2.5. Self-Esteem

Self-esteem was assessed by the 10-item Rosenberg Self-Esteem Scale (RSES; [57]), which is rated on a 4-point Likert scale, ranging from 0 (strongly disagree) to 3 (strongly agree). RSES was designed to assess overall feelings of self-importance and self-worth in individuals (e.g., "I feel that I have a number of good qualities"; "I take a positive attitude toward myself"). A higher score indicates a greater level of self-esteem. The RSES has shown good internal consistency in samples of university students, e.g., (Cronbach's $\alpha \geq 0.81$; [58,59]). In the present study, the RSES demonstrated good reliability (Cronbach's $\alpha$ from 0.84 to 0.93).

### 2.3. Procedure

The present study was carried out as part of the COST Action Appearance Matters IS1210. Ethics approval was obtained from the relevant bodies in all participating countries: Croatia (Ref. Nr.: 11-73/16-2231), Finland (Ref. Nr.: 38/2016), Ireland (Ref. Nr.: HS-16-69), Israel (Ref. Nr.: 2988, 8/12/2016), Portugal (Ref. Nr.: 1-12/2016), Bosnia and Herzegovina, Belgium, and Greece (approval without Ref. Nr.).

The questionnaires which were not available in the language of participating countries were translated into the respective languages following the Translation, Review, Adjudication, and Documentation process (TRAPD; [60]).

To approach the largest possible group of students, invitations to complete the survey were sent by individual universities. The students received an email through their institutional account containing a brief description of the study and a link to the secure online survey. In some universities, participants were also recruited via the student union and its suborganizations and the intranet for students (information channel intended for all students). Participants were informed online about the ethical principles of the study (voluntary participation, protecting the confidentiality of data, the right to withdraw from the research at any time, and principal investigator contact). After providing digital informed consent, participants were asked to complete the instruments described above.

### 2.4. Data Analyses

No missing data were found, and outliers were eliminated. The analytic strategy included preliminary analysis to ensure the equivalence of measures to proceed with cross-cultural research. These analyses are described in detail in the Supplementary Material. Briefly, a confirmatory factor analysis (CFA) was conducted to examine whether the proposed one-factor structure of the BAS-2 adequately represented our data on the total sample (Supplemental Figure S1). We also assessed the measurement invariance of the BAS-2 model in different countries' subsamples, performing the Multi-Group Confirmatory Factor Analysis (MGCFA; Supplemental Table S1). According to the MGCFA results, the configural and metric invariance were met, but scalar invariance was weak. Therefore, mean differences in BAS scores across countries need to be interpreted considering possible differences in the meaning of the latent construct in different countries. The CFA was also conducted to check the proposed one-factor structures of the SRSP, CAS-R/Ed, and RSES (Supplemental Table S2). The findings confirmed the use of the scales' total scores in correlational analyses for the total sample.

To address the study aims, we conducted a one-way between-groups analysis of variance (ANOVA) to explore the differences in PBI between countries. Eta-squared ($\eta^2$) was interpreted according to the commonly used Cohen's [61] guidelines: 0.01 = small effect; 0.06 = moderate effect; 0.14 = large effect. The effects of BMI on PBI were calculated on a reduced sample (n = 1989), as in Belgium, Finland, Israel, and Ireland, no data on BMI were collected. Pearson correlation coefficients (*r*) were calculated to explore the relationship between BMI and BAS-2 total scores. Next, we regressed BAS-2 scores on BMI to obtain unstandardized regression coefficients, standard errors of coefficients, and related confidence intervals for testing the moderation effect of the country on the BMI–BAS relationship. In addition, we analyzed the BMI–BAS relationship based on the weight categorization of the participants. Participants were divided into three weight categories based on the World Health Organization's BMI standards [62]: the underweight (UW) category included 9.5% of the participants, the normal weight (NW) category 75.5%, and the overweight/obesity (OW/OB) category encompassed 15.0% of the sample. Due to differences in sample size across categories, the UW category could not be included in the one-way ANOVA conducted to compare weight categories regarding PBI. To test the interaction effect of the BMI and country categorization on PBI, we conducted a 2 (NW, OW/OB) × 4 (Bosnia and Herzegovina, Croatia, Greece, and Portugal) two-way ANOVA. Finally, we analyzed correlations between study variables and tested the hypothesized mediation effect of self-esteem on the relation between PBI and academic outcomes.

AMOS 20 was used for data analysis. The alpha level was set at 0.05.

## 3. Results

### 3.1. Country Differences in Positive Body Image

Differences between countries on the BAS-2 mean scores were tested by the ANOVA model. The assumption of homogeneity of error variance was tested by Levene's test, $F(7, 2642) = 2.57$, $p = 0.01$, indicating reasonably equal error variances between groups. The highest average PBI was reported by the female students in Bosnia and Herzegovina ($M = 3.89$, $SD = 0.70$), and the lowest was in Ireland ($M = 3.16$, $SD = 0.81$; see Table 1). There was a significant difference between countries in PBI, $F(7, 2642) = 18.84$, $p < 0.001$, $\eta^2 = 0.048$, $p < 0.01$, and Scheffe post hoc comparisons, represented as homogeneous subsets in Table 1, revealed these differences. Summarizing these multiple comparisons, it can be concluded that the highest body appreciation was reported in Bosnia and Herzegovina, Croatia, and Israel; the medium level in Greece and Portugal; and the lowest level of body appreciation was found in Belgium, Finland, and Ireland. However, the effect size indicates that differences attributable to the country of residence on BAS-2 are small.

**Table 1.** Descriptive Statistics and Group Differences in BAS-2 Total Scores by Countries—F-test and Post hoc Analysis (Scheffe—Homogenous Subsets).

| Country | *n* | Subset *M (SD)* | | | | |
|---|---|---|---|---|---|---|
| | | **1** | **2** | **3** | **4** | **5** |
| Ireland | 88 | 3.16 (0.81) | | | | |
| Finland | 263 | 3.35 (0.74) | 3.35 (0.74) | | | |
| Belgium | 135 | 3.43 (0.67) | 3.43 (0.67) | 3.43 (0.67) | | |
| Portugal | 517 | | 3.58 (0.77) | 3.58 (0.77) | 3.58 (0.77) | |
| Greece | 1040 | | | 3.66 (0.74) | 3.66 (0.74) | 3.66 (0.74) |
| Israel | 158 | | | 3.71 (0.87) | 3.71 (0.87) | 3.71 (0.87) |
| Croatia | 201 | | | | 3.83 (0.68) | 3.83 (0.68) |
| B&H | 248 | | | | | 3.89 (0.70) |
| *p* | | 0.058 | 0.205 | 0.055 | 0.154 | 0.257 |

*Note*: B&H = Bosnia and Herzegovina. The post hoc differences between countries in different subsets are significant at $p < 0.05$ level.

### 3.2. The Effects of BMI on Positive Body Image

The correlation between BMI and BAS-2 was negative and low-to-moderate (see Table 2). On the total sample, $r = -0.33$, and within the countries, it varied from $r = -0.26$ in Bosnia and Herzegovina to $r = -0.36$ in Greece.

**Table 2.** Correlations Between BMI and Positive Body Image (BAS-2).

| | *n* | *r* | $R^2$ | *b* | $se_b$ | 95% CI for *b* | |
|---|---|---|---|---|---|---|---|
| | | | | | | **LB** | **UB** |
| Total sample | 1989 | −0.33 | 0.109 | −0.071 | 0.005 | 0.080 | 0.062 |
| B&H | 243 | −0.26 | 0.068 | −0.077 | 0.018 | −0.113 | −0.041 |
| Croatia | 199 | −0.34 | 0.116 | −0.076 | 0.015 | −0.106 | −0.047 |
| Greece | 1030 | −0.36 | 0.130 | −0.072 | 0.006 | −0.083 | −0.061 |
| Portugal | 517 | −0.27 | 0.073 | −0.061 | 0.010 | −0.079 | −0.042 |

*Note*. B&H = Bosnia and Herzegovina; CI = confidence interval; *LB* = lower bound; *UB* = upper bound.

The moderation effect of the country on BMI–BAS relation was not observed, the correlation and regression coefficients were similar across countries, and 95% confidence intervals were largely overlapping. The percentage of variance accounted for by BMI in body appreciation was around 11% in the whole sample and ranged from 7% in Bosnia and Herzegovina to 13% in Greece.

The second approach to analyzing the BMI–BAS relationship was based on the weight categorization of the participants. The ANOVA results supported the previous correlational findings showing the difference in PBI between OW/OB and NW groups, $F(1, 1792) = 87.39$, $p < 0.001$, $\eta^2 = 0.05$. The average BAS score in the NW group ($M = 3.76$, $SD = 0.71$) was higher than in the OW/OB group ($M = 3.21$, $SD = 0.79$). The interaction effect of BMI category and country of residence on body image was non-significant, $F(3, 1792) = 0.10$, $p = 0.96$, $\eta^2 = 0.01$, indicating that the effect of BMI on body appreciation was not observed to differ across countries.

### 3.3. Relations between Positive Body Image, Self-Esteem, Perceived Academic Achievement, and Educational Aspirations

The correlation between BAS-2 and self-esteem was high ($r = 0.69$; Table 3). These two variables correlated significantly with both variables of the educational domain ($p < 0.01$), with BAS-2 presenting a higher correlation with perceived academic achievement ($r = 0.29$) and self-esteem showing a higher correlation with educational aspirations ($r = 0.40$).

**Table 3.** Correlation Between Positive Body Image, Self-Esteem, Perceived Academic Achievement, and Educational Aspirations (N = 2635).

| Measure | BAS-2 | RSES | SRSP |
|---|---|---|---|
| RSES | 0.69 *** | | |
| SRSP | 0.29 *** | 0.25 *** | |
| CAS-R/Ed | 0.19 *** | 0.40 *** | 0.29 *** |

*Note.* BAS-2 = Body Appreciation Scale-2; RSES = Rosenberg Self-Esteem Scale; SRSP = Self-Reported Study Performance; CAS-R/Ed = Educational Aspiration subscale of the Career Aspiration Scale-Revised. *** $p < 0.001$.

We also tested the hypothesized mediation effect of self-esteem on the relationship between BAS-2 and perceived academic achievement (see Figure 1) and the relation between BAS-2 and educational aspirations (see Figure 2).

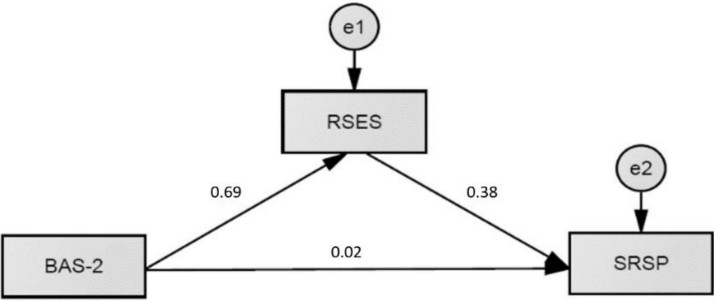

**Figure 1.** A Mediation Model of Self-Esteem (RSES) on the Relation Between Positive Body Image (BAS-2) and Perceived Academic Achievement (SRSP); Standardized Estimates Are Shown.

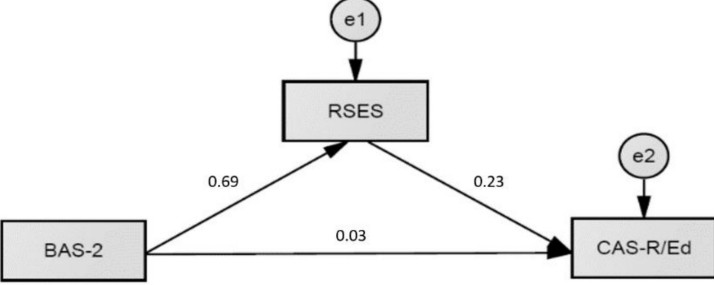

**Figure 2.** A Mediation Model of Self-Esteem (RSES) on the Relation Between Positive Body Image (BAS-2) and Educational Aspirations (CAS-R/Ed); Standardized Estimates Are Shown.

Self-esteem fully mediated the relationship between BAS-2 and perceived academic achievement (standardized path c′ = 0.02; $p = 0.95$), with a significant indirect path BAS-2

→ RSES → SRSP (standardized path axb = 0.27; *p* < 0.001). A similar mediation effect of self-esteem was shown in the association between PBI and educational aspirations. Self-esteem fully mediated this relationship (standardized path c′ = 0.03; *p* = 0.95) with a significant indirect path BAS-2 → RSES → CAS-R/Ed (standardized path axb = 0.16; *p* < 0.001).

### 3.4. The Full Model of Body Image Effects on Perceived Academic Achievement and Aspirations

Considering the previously presented results, including the relationship between BMI and PBI, we tested a general model of body image effects on perceived academic achievement and aspirations. This model was tested on a reduced sample size due to missing data of BMI in some countries (*N* = 1,989). The proposed model had a moderate fit to the data ($\chi^2$ = 52.5, $\chi^2/df$ = 10.50, CFI = 0.977, NFI = 0.975, RMSEA = 0.069, 90% CI [0.053, 0.087]). After consulting the modification indices, we proposed an additional bivariate relation between BMI and self-esteem (Figure 3).

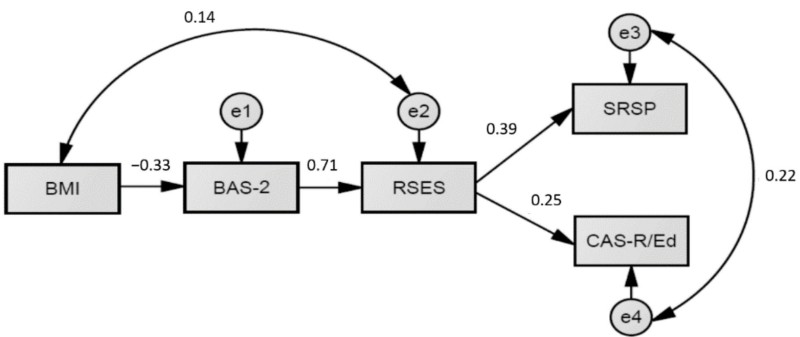

**Figure 3.** The modified full model of positive body image effects on academic performance and aspirations. *Note.* BMI = Body Mass Index; BAS-2 = Body Appreciation Scale-2; RSES = Rosenberg Self-Esteem Scale; SRSP = Self-Reported Study Performance; CAS-R/Ed = Educational Aspiration subscale of the Career Aspiration Scale-Revised; standardized estimates are shown.

The modified model had an excellent fit to the data ($\chi^2$ = 14.9, $\chi^2/df$ = 3.73, CFI = 0.995, NFI = 0.973, RMSEA = 0.037, 90% CI [0.018, 0.058]). Therefore, it can be inferred that BMI correlated with self-esteem residual (error) variance, despite its indirect effect via PBI. Although the model demonstrated a good fit to the data, the variance accounted for in perceived academic achievement ($R^2$ = 0.07) and educational aspirations ($R^2$ = 0.16) by body image was low but statistically significant, indicating the value of testing these variables. Furthermore, it should be noticed that PBI affected educational variables only indirectly through the mediating effect of self-esteem.

## 4. Discussion

The present study aimed to investigate the relationship between PBI and perceived academic achievement and educational aspirations in female university students from an international perspective. This work elucidated the hypothesized mediation effect of self-esteem on the relationship between these variables.

The study results also confirmed the one-factor structure of BAS-2 as a quality PBI measure for female young adults, as well as its reasonable cross-cultural structural invariance, mirroring results from past research, e.g., [37,63–65]. Previous research [66] has highlighted the lack of cross-cultural validity on body image measures, such as the BAS-2, due to a lack of multi-group confirmatory factor analyses (MGCFA) with different language groups, and this gap was addressed in the current study. Some small effect size differences in body appreciation between countries were found. The highest PBI was reported in Bosnia and Herzegovina, Croatia, and Israel; medium level in Greece and Portugal; and the lowest levels of PBI were found in Belgium, Finland, and Ireland. The possible sources of these differences may be attributed to cultural differences between these clusters of countries. Even considering the spread of thin-ideal associated with the globalization

phenomenon, beauty ideals can differ slightly across countries. Indeed, one study found that Bosnian females reported their country's ideal body size as bigger than their Western counterparts [67]. Given the negative relationship between the internalization of thin-ideal and body appreciation [5], women living in countries where ideal bodies are more realistic, and beauty is broadly defined are more likely to accept their own bodies. Moreover, some PBI components, such as adaptive appearance investment [68] and appreciation of body characteristics [65], can be differently perceived across national groups, generating subtle variations in PBI. It should be noted that differences among countries were of small magnitude, and the presumption of scalar invariance was not fully met. Thus, these differences should be interpreted with caution. Similarly, Lemoine and colleagues [37] found marginal differences between the three European countries as well as a small effect size. Therefore, we can assume that European countries with similar cultural background demonstrate relatively similar levels of body appreciation, even though more research is needed to draw solid conclusions.

We found a negative and moderate relationship between BMI and PBI, and this relationship was stable across countries. Accordingly, OW/OB participants had lower PBI scores than NW participants in all national groups. These findings align with previous studies performed with female university students [38,45,58,69] and emerging adults [61] and lend credence to the view that body appreciation can be influenced by BMI but is not a function of it. Attitudes towards weight can influence body appreciation [41] but are not the only determining factor, substantiating the multifaceted nature of PBI [35].

In terms of PBI correlates, we found that higher self-esteem and educational aspirations, as well as students' perception of greater academic success, were associated with increased body appreciation. The link between PBI and self-esteem in university students is substantiated by existing research, e.g., [38,45,69], and gives strength to the notion that appearance is one important source of self-esteem in this population [70]. From a theoretical point of view, some PBI features have the potential to protect, and even strengthen, global self-esteem, in particular the ability to resist the internalization of media influences, to accept one's own body, and to feel beautiful and worthiness on the inside [36]. Body acceptance means that an individual feels comfortable with their body even if not completely satisfied with all physical aspects [35], and this way to cope with undesired features may be protective to self-esteem. Lastly, there is also a possibility that body confidence can extend to a general feeling of confidence regarding other capabilities.

Based on the magnitude of the relationships, we also conclude that PBI is more strongly associated with perceived academic achievement (moderate correlation) than educational aspirations (low correlation). Nevertheless, according to the results of the mediation models tested in this study, the effect of PBI on these academic outcomes seems to occur via self-esteem. In this process, body weight does not have a direct influence on academic variables; it is rather a risk factor for poor body image and self-esteem.

The mediating role of self-esteem suggests that benefits in perceived academic achievement and aspirations can be derived from positive self-image beyond the PBI component. Self-esteem is a comprehensive construct that refers to an overall sense of self-worth. As it serves a motivational function, self-esteem can leverage motivation to succeed in academic setting by implementing behaviors that promote better outcomes [52] and developing coping skills to deal with overcoming adversities [71]. Confidence in one's capabilities can also be a basis for engaging in more demanding activities, such as planning longer-term educational goals. Fear of failure may cause students with low self-esteem to hold back, whereas students with high self-esteem may be more likely to feel challenged and, in this case, more motivated to pursue an advanced education program.

It needs to be acknowledged that the total effects on academic outcomes were small in size; despite this, they enable us to conclude that body image can (indirectly) play some role in academic outcomes. Low effects indicate the need for future studies to test potential moderators and additional mediators. Self-efficacy, academic self-concept, locus

of control, and motivation for academic success are possible mitigating variables that must be examined.

## 5. Strengths and Limitations

To our knowledge, this was the first study analyzing the link between PBI and perceived academic achievement and educational aspirations, emphasizing the relevance of self-esteem. Another strength of this study was its cross-national design, allowing a comparison between countries and a formulation of a comprehensive model for several national samples. Moreover, we have focused on university students, differently to past research that was mostly carried out with adolescents. Distinct interactions between these variables are plausible depending on the developmental phase.

Nevertheless, some limitations must be acknowledged. First, this study was based on a cross-sectional design and therefore did not allow testing temporal precedence and causality between variables. For this reason, alternative pathways are also conceivable, including the possibility of perceived academic achievement and educational aspirations impacting PBI [25] and/or self-esteem [43]. Additionally, the relationship between self-esteem and PBI could be conceptualized differently, namely by hypothesizing the existence of reciprocal influences [72]. Thus, longitudinal studies are required to validate the full theoretical model proposed in this study. Second, our sample consisted of female participants only, which limits the generalization of findings to the population of male and other gender university students. Male and non-binary students are also important samples to study. Future investigations should be open to formulating different models for such samples. As an example, Park [72] found that the causal direction from body image to self-esteem was not significant for teenage boys. Second, BMI was not collected in Belgium, Finland, Israel, and Ireland, and the analysis of UW students was limited due to the small sample size. In addition, BMI was calculated based on self-report, which can result in inaccuracies of actual weight and height. Third, future research should employ a more comprehensive assessment of the study variables. For example, with regard to educational aspirations, dimensions such as leadership aspirations and achievement aspirations are relevant and should be considered in future studies. It is also suggested that future studies provide a deeper understanding of the PBI construct beyond the information provided by the BAS-2. For instance, body functionality and inner positivity are other PBI facets pertinent to investigate due to their potential to promote psychological well-being [73].

## 6. Conclusions

This study leveraged a positive psychology approach to examine how body appreciation and self-esteem relate to perceived academic achievement and aspirations. Although the mediation analyses do not allow us to draw causal deductions, the outcomes denote that the co-occurrence of PBI and high self-esteem create favorable conditions for academic success. Although body appreciation did not directly influence academic outcomes, practitioners should not ignore the potential benefits of PBI, as students with higher PBI are likely to have higher self-esteem. Thus, programs offered at colleges and universities should aim to enhance these aspects of psychological health on the assumption that they are tightly interwoven and could facilitate students' academic outcomes and overall well-being. The accumulation of positive psychological resources may not only help individuals to enhance their performance in academic setting [74] but also protect them against pathology [75]. By exploring individual inner qualities and strengths, the risk of students developing disorders that impact their academic career (such as depression, anxiety, and eating disorders) could be minimized.

**Supplementary Materials:** The following are available online at https://www.mdpi.com/article/10.3390/ejihpe12100107/s1, Figure S1: Structural model of the BAS-2 with standardized estimates obtained in the complete sample, Table S1: Measurement Invariance of the BAS-2: Model Fit Indices for the Multigroup Models—Country Invariance, Table S2: CFA Results for the Self-Reported Study

Performance Scale, Educational Aspiration Scale, and Rosenberg Self-Esteem Scale on the Total Sample [76–79].

**Author Contributions:** Conceptualization, S.T., C.A.A., A.F., B.D., A.L., C.M.C., D.O.-S., E.S., J.B., L.B., M.Š.-M., M.V., M.A., T.H. and T.B.; Methodology, S.T., C.A.A., A.F., B.D., A.L., C.M.C., D.O.-S., E.S., J.B., L.B., M.Š.-M., M.V., M.A., T.H. and T.B.; Formal Analysis, T.B., S.T. and C.A.A.; Investigation, S.T., C.A.A., A.F., B.D., A.L., C.M.C., D.O.-S., E.S., J.B., L.B., M.Š.-M., M.V., M.A., T.H. and T.B.; Resources, S.T., C.A.A., A.F., B.D., A.L., C.M.C., D.O.-S., E.S., J.B., L.B., M.Š.-M., M.V., M.A., T.H. and T.B.; Writing—Original Draft Preparation, S.T., C.A.A. and T.B.; Writing—Review and Editing, B.D. and A.F.; Project Administration, T.B. All authors have read and agreed to the published version of the manuscript.

**Funding:** This research was funded by the Center for Psychology at the University of Porto, Portuguese Science and Technology Foundation, FCT UIDB/00050/2020, FCT UIDP/00050/2020.

**Institutional Review Board Statement:** The study was conducted according to the guidelines of the Declaration of Helsinki, and ethic approval was obtained from the relevant bodies in all participating countries: Croatia (Ivo Pilar Institute of Social Sciences, Ref. Nr.: 11-73/16-2231); Finland (University of Turku, Ref. Nr.: 38/2016); Ireland (University College Dublin, Ref. Nr.: HS-16-69); Israel (The Open University of Israel, Ref. Nr.: 2988, 8/12/2016); Portugal (University of Porto, Ref. Nr.: 1-12/2016); Bosnia and Herzegovina (University of Mostar), Belgium (Ghent University), and Greece (Aristotle University of Thessaloniki) approval without Ref. Nr.

**Informed Consent Statement:** Digital informed consent was obtained from all subjects involved in the study.

**Data Availability Statement:** The data that support the findings of this study are available from the corresponding author (S.T.) upon reasonable request.

**Acknowledgments:** This article is based upon work from COST Action Appearance Matters IS1210, supported by COST (European Cooperation in Science and Technology). www.cost.eu.

**Conflicts of Interest:** The authors declare no conflict of interest.

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
