# Peer review of "An International Study of Correlates of Women’s Positive Body Image"

_ejihpe, doi:10.3390/ejihpe12100107_

Round 1

Reviewer 1 Report

An appropriate conclusion in clinical terms is not developed. Please note: the paper in some sections suggests that positive body self-esteem is associated with low BMI, and success. It is important to understand if you are finding that success is associated with good body self-esteem (regardless of BMI). We need to discuss the clinical implications by stressing the positive psychology that the authors cite. Success and aspirations should NOT depend on body weight. If so, in all these nations an emormous work should be done on body acceptance, body positivity and success in life INDEPENDENTLY on weight.

The study present as positive the link between low BMI and success.

I believe this can be a dangerous interpretation. Success and aspiration among young women should be independent from body mass index and weight. Consequently the fact that was found this association is rather a risk factor.

I reccomend a review of the results comments and interpretations.

Reviewer 2 Report

Manuscript: ejihpe-1831049

I find the manuscript interesting and well presented. The inclusion of data from eight European countries make the results easier to generalize. The different sections in the manuscript are well performed, the statistical analyses are adequate to the study´s aim, and the discussion is well developed. I consider a strength the authors' effort to seek cross-cultural structural invariance in the scales used.

 However, the most important concern, from my point of view, is the place self-esteem should have.  Authors are aware of this when they write in lines 427-429 (Strengths and Limitations section) “Also, the relationship between self-esteem and PBI could be conceptualized differently, namely hypothesizing the existence of reciprocal influences”. It is more usual to assume that high academic achievement and educational aspirations will increase self-esteem, whereas, the opposite, that high self-esteem will increase academic achievement and educational aspirations will always need other factors such as time to study, motivation, etc…. I think that authors should accentuate why the direction in the study is that self-esteem is the mediator variable and not the dependent one.

Smaller concerns are listed in the next paragraphs:

1. The use of a couple of items to describe the scales used should extended to all scales, including BAS-2 and RSES.

2. Although all variables in the study are obtained from self-reports, I think that in the case of the “Academic Achievement” variable, the use of the word “perceived” should be introduced, because it is not an objective achievement, but a “perceived academic achievement”. My advice is to change the name of the variable throughout the manuscript.

3. The reliability (Cronbach´s Alphas) in the study are in a supplementary file, however the reliability from other studies are included in the manuscript (Materials and Methods section, measures subsection). Authors should include the measures´ reliability of the study in the text, and delete this part from the supplementary file. Also, when introducing the reliability, because of the presence of values from the eight countries, I suggest the authors use the maximum and minimum values, i.e. BAS-2 scale (reliability from .91 to .96). It is more important the reliability of this study than the reliability of others.

4. In some countries there is no information about BMI (Belgium, Finland, Israel and Ireland). This should be included as a limitation.

5.  Authors have chosen to ignore the underweight (UW) category (9.5% of the participants) in the ANOVA analysis (because the small sample size). However, the non-parametric statistical analyses can be used in this kind of situations. In my opinion, the study would be more complete if a non-parametric analysis such as Kruskal-Wallis was performed.

6. I find all Tables in the manuscript too close to the text, if possible separate them from the text.

7. Table 3. All correlations are significant at p<.01 (including correlations as different as .19 and .69). My advice is the use of a significance level of p<.001.

8. Figure 1A, should be split in two figures to increase the size of the standardized estimates, (they are very difficult to see). And they should include the significant level for each of them.

Round 2

Reviewer 1 Report

I believe the authors clarified the resulta and the association found.